# Design Proposal for Sign Language Services in TV Broadcasting from the Perspective of People Who Are Deaf or Hard of Hearing

**Ji Hyun Yi [1], Songei Kim [1], Yeo-Gyeong Noh [1], Subin Ok [1] and Jin-Hyuk Hong [1,2,*]**

[1] School of Integrated Technology, Gwangju Institute of Science and Technology, Gwangju 61005, Korea; doghyun@gist.ac.kr (J.H.Y.); xong2222@gist.ac.kr (S.K.); ygnoh0210@gist.ac.kr (Y.-G.N.); subinok130@gm.gist.ac.kr (S.O.)
[2] Artificial Intelligence Graduate School, Gwangju Institute of Science and Technology, Gwangju 61005, Korea
* Correspondence: jh7.hong@gist.ac.kr

**Abstract:** Sign language services are provided so that people with hearing loss are not alienated from socially and politically important information through TV broadcasting. In this paper, we conducted a user survey and evaluation of the current sign language services for deaf or hard-of-hearing (DHH) people, and solutions were proposed for the problems found in the course of the analyses. To this end, a total of five stages of research were conducted. First, the communication problems experienced by DHH individuals and previous studies on their language and information acquisition were investigated. Second, the most typical types of information delivery channels via TV were defined as news, discussions, and weather reports, and by investigating the actual sign language service cases for each type, three visual information delivery elements were identified: sign language interpreters, reference videos, and subtitles. Third, a preference survey, an interview survey, and an eye tracker experiment on the DHH participants were conducted with varying arrangement options of information delivery elements. Fourth, based on the results of the investigations and experiments, the options to be considered when arranging information delivery elements were compiled. The results showed that the sign language interpreter, which is the first element of information delivery, should be presented in a size clearly visible because the visibility of their facial expressions is important. In addition, it is recommended to present the interpreter without a background since DHH participants did not prefer the presence of a background. As for subtitles, which is the third element of information delivery, it was confirmed that the provision of sign language interpretation and subtitles together helped DHH participants to understand the contents more quickly and accurately. Moreover, if there are multiple speakers, individual subtitles for each speaker should be provided so that the viewers can understand who is talking. Reference videos, which are mainly placed on the screen background, the second information delivery element, were considered less important to DHH participants compared to sign language interpreters and subtitles, and it was found that DHH participants preferred reference videos to be visually separated from sign language interpreters. Fifth, based on the overall results of the study, a screen layout design was proposed for each type of information delivery element for DHH people. Contrary to the general conception that there would be no problem in viewing information-delivering TV broadcasts by DHH people simply by placing a sign language interpreter on the screen, the results of this study confirmed that a more delicate screen layout design is necessary for DHH people. It is expected that this study will serve as a helpful guide in providing better sign language services for TV broadcasts that can be conveniently viewed by both DHH and non-disabled people.

**Keywords:** deaf or hard-of-hearing (DHH) people; TV sign language service; usability test; layout design

## 1. Introduction

TV broadcasting, which is typically accessible to everyone, is an important medium for delivering crucial information, such as important news and government policy announcements, to the public. During the ongoing battle against COVID-19, we have often needed to listen to government briefings about the virus. When delivering such important information, a sign language interpreter is present next to the speaker so that DHH people can also understand the information. In 2020, the U.S. government was sued by the DHH association for not providing sign language interpreters during a COVID-19 briefing in the U.S. [1].

It is recognized that sign language should be provided as a basic service when delivering socially and politically important information. As shown in Figure 1, Korean sign language services are currently provided during the government's COVID-19 briefings or major news sessions by major broadcasters in Korea. Although Korean sign language interpreters appear along with the speaker in the government's COVID-19 briefings [2], they are usually placed in a circle or square in the lower right corner of the screen with the size of about a quarter of the vertical length of the screen in usual TV news programs [3]. These screens are viewed by both people with normal hearing and those who are DHH together. As shown in Figure 1b, a Korean sign language interpreter is usually presented in the lower right corner in a very small size, which leads to a question of whether their hands, a crucial means of communication in sign language, are sufficiently recognizable to DHH people. In situations where sign language services are becoming the norm, the research team of this study attempted to identify and solve the potential problems lying in the current sign language services from the perspective of DHH people.

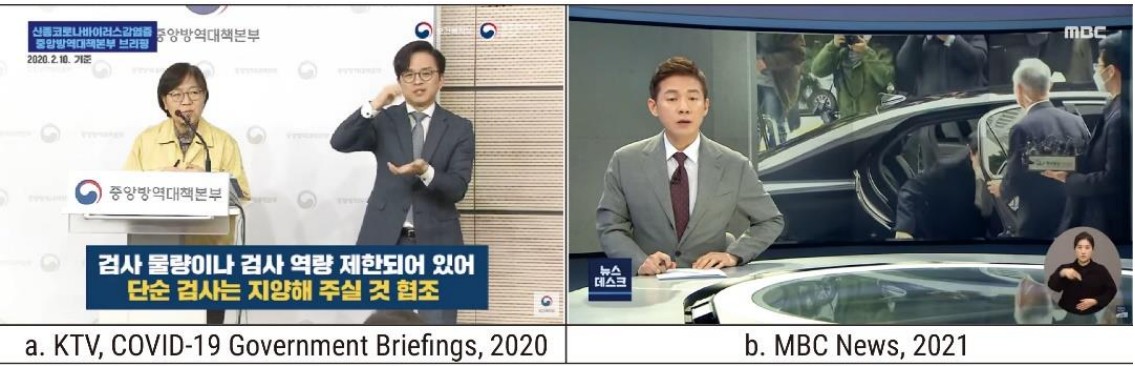

**Figure 1.** Samples of sign language services in TV broadcasting in Korea. Translation of subtitle–Left: "Please refrain from simple inspections due to limited inspection volume and inspection capabilities".

DHH individuals do not have a visibly distinguishable disability compared to others; what they suffer is problems due to their difficulties in hearing. The cognitive ability of DHH people is not different from that of non-disabled people [4,5]. However, Dyer et al. stated through several research examples [6–8], "In most deaf people, reading and writing skills fail to achieve levels appropriate to the age and intelligence of the student, typically lagging their peers by several years in the final years of obligatory schooling". In addition, he mentioned that the reason for their low functional literacy was because most DHH people have restricted access to spoken language forms [9]. This is because there is a big difference between acquiring a language through hearing and acquiring a language through sight. Because of the loss of hearing, it is difficult for DHH people to learn language in a similar way to non-disabled people [10]. Auer and Bernsteins suggested through experiments that "the acquisition of a speech vocabulary available by eye alone not only is smaller but also shows different distributional characteristics than that for heard speech" [11]. Thus, because DHH people rely on sight to acquire information, their literacy ability is significantly lower than non-disabled people [9,12].

With regards to the language and information acquisition of DHH people, Korean Sign Language [13], a study on information accessibility of the disabled [14–16], a study on subtitles and visualization for DHH people [17–21], etc., and research on sign language interpretation screens using eye trackers have also been conducted [22,23]. These studies have suggested diverse aspects related to the language and information acquisition of DHH people. First, the language used by non-disabled people is different from sign language in terms of grammar and word order. Therefore, the subtitles displayed following the grammar and word order of the language used by non-disabled people would be difficult to understand for those DHH people whose mother language is a sign language. Second, there are many DHH people whose literacy or ability to communicate in sign language is poor due to lack of proper sign language education. It is therefore important to give DHH people equal opportunities in terms of easy access to information acquisition via TV broadcasts. Through sign language services that provide information at a level comprehensible by non-disabled people as well, DHH people will be given the opportunity to understand and be exposed to grammar, word order, and various words of the language used by DHH people. Sign language services in TV broadcasts not only provide opportunities for DHH individuals, as citizens, not to be alienated from important information but also empowers them to improve their language skills. This will have the effect of giving a practice opportunity to match the language of non-disabled people with the language of DHH people.

DHH people who have lost hearing rely mostly on sight to receive information. In order to allow DHH people to better acquire information from TV broadcasts, the arrangement of visual elements is important for delivering information to them.

In Korea, the Korea Communications Commission and Telecommunications Technology Association have provided Korea Smart Sign Language Broadcasting Services as personalized services for DHH people since 2020. This service can be viewed on major channels via IPTV. Through these services, it is possible for DHH individuals to watch TV programs with adjustable screen size, location, subtitles, etc., for the convenience of information acquisition, as shown in Figure 2. These Korean sign language broadcasts allow DHH individuals to freely turn on and off the sign language screen and adjust the size and position of the screen as well. In order to use the smart sign language broadcasting service, however, a TV that supports such functions is required. In normal TV sets without the required functions, it is impossible to adjust the screen elements for the sign language.

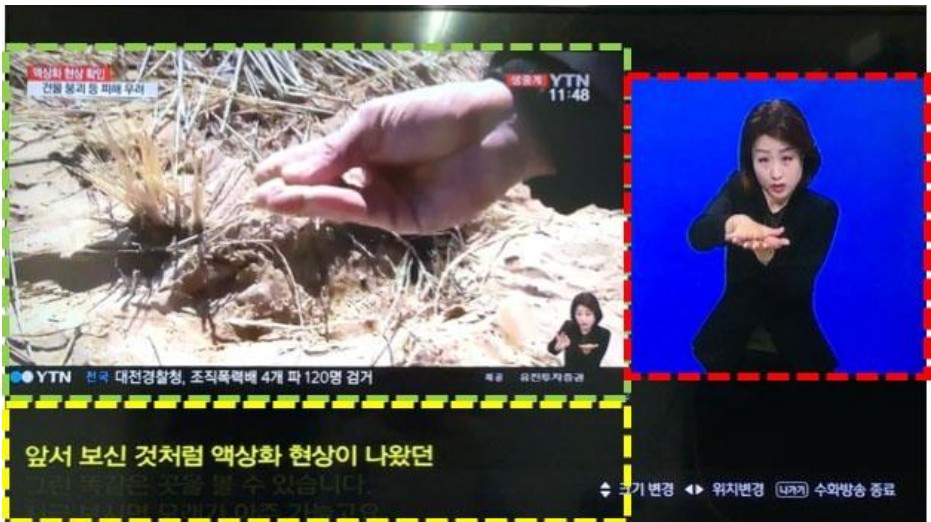

**Figure 2.** Example of smart Korean sign language broadcasting services. Translation of subtitle–In yellow box: "As you saw earlier, the liquidation phenomenon came out".

In this study, we tried to examine the arrangement of information for DHH people on the screen of ordinary TVs not supporting screen element adjusting controls. First, user experiments were conducted with various element placement conditions for DHH individuals to define the characteristics of information delivery types and major visual elements for information delivery for DHH individuals, and to find out how to arrange these elements so that both DHH and non-disabled people can acquire information efficiently. Based on the results, screen arrangement options for visual information delivery elements for DHH people were proposed.

## 2. Visual Information Delivery Elements for DHH People to Understand Information from TV Broadcasts

We investigated actual cases of information delivery for DHH people using news programs: media that delivers essential information covering political, social, and economic affairs while excluding dramas and other entertainment programs. In Korea, sign language interpretation services are being provided to deliver information such as government policy announcements and important news. Three types of information delivery channels, such as news, panel discussions, and weather forecasts, were selected for the analyses of this study. Each of the visual features appearing in these types and cases when sign language service is added were investigated and analyzed.

### 2.1. Characteristics of Sign Language Service Screen Arrangement by Type of Information Delivery-Oriented TV Broadcasting

2.1.1. News

News conveys information about social, economic, and cultural events and incidents. As shown in Figure 3, one main reporter delivers information about multiple news pieces, and journalists explain the pieces with reference video either by appearing in the screen in person or only with narration. In general, short headlines or subtitles that sum up the key information are provided on the screen as well.

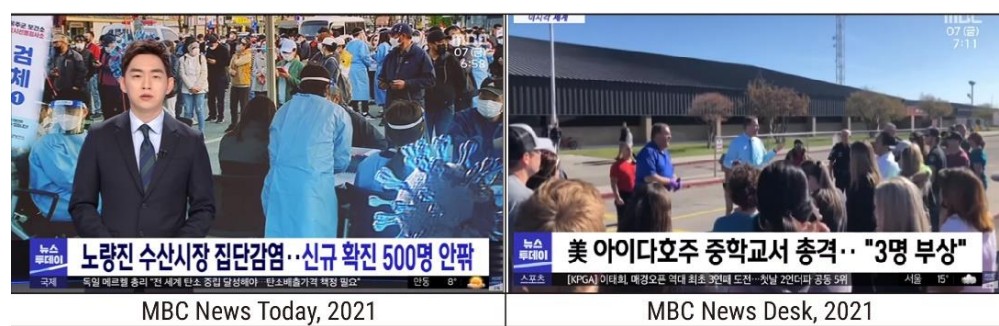

**Figure 3.** Example of news type broadcasts. Translation of subtitles–Left: "Group infection at Noryangjin Fish Market . . . Around 500 new confirmed cases". Right: "Shooting at a middle school in Idaho . . . 3 people were injured".

When Korean sign language services are provided, the data image is placed like a background, and the sign language interpreter appears small in the lower right corner of the screen, as shown in Figure 4. In the case of major government policy briefings, a sign language interpreter appears and interprets. This is sometimes provided on a screen separate from the main video, as in the example of the United States (see the rightmost example in Figure 4).

2.1.2. Panel Discussion

As seen in Figure 5, multiple speakers exchange their opinions in panel discussion settings. Reference videos or subtitles are generally not provided in panel discussions. In these settings, several people sometimes speak at the same time and for non-disabled people, it is not difficult to understand who is speaking and who is not, since they can

distinguish speakers' voices and the movement of their lips. However, for DHH people, it can be hard to identify the speaker only by the subtitles provided. In a discussion, not only the information delivered by the speaker's comment but also the emotion conveyed can play an important role; however, the current sign language and subtitles system cannot deliver emotional information. Moreover, in the case of TV programs dealing with current affairs and political debates, complicated words that are not frequently used appear frequently, making it even harder to convey the contents through sign language.

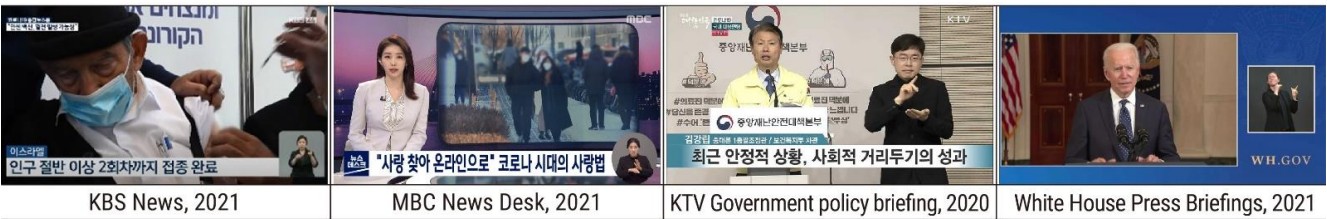

**Figure 4.** Example of sign language services in news.

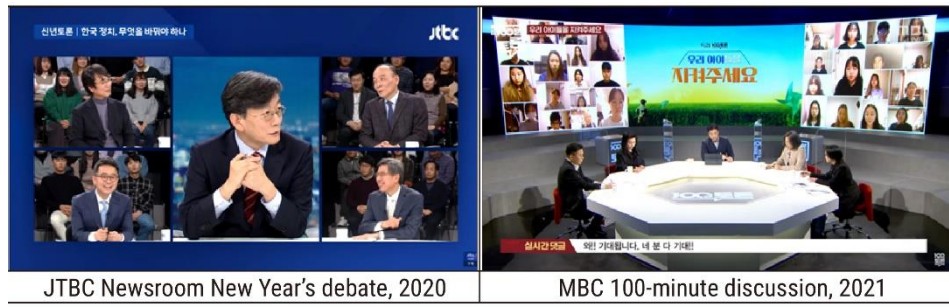

**Figure 5.** Examples of discussion type broadcasting.

Looking at the case where the sign language service is provided for panel discussions, one Korean sign language interpreter interprets for all the speakers, as shown in the left example of Figure 6. In this case, it is not easy to tell who is currently speaking, as discussed above. On the other hand, in some cases, as seen in the photo on the right side, there are individual American sign language interpreters for the moderator and two speakers, so it is possible to distinguish who is speaking. For this to happen, multiple sign language interpreters are required.

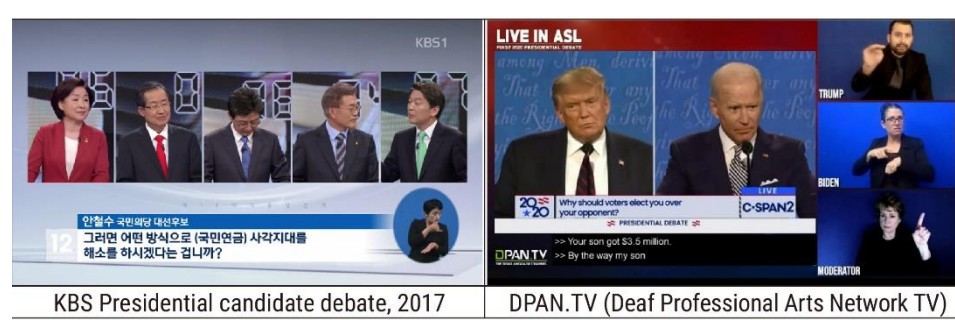

**Figure 6.** Example of sign language services in discussion.

### 2.1.3. Weather Broadcasts

In the case of weather forecasts, as shown in Figure 7, graphics and figures for weather information are placed on the entire background screen, and a weather caster explaining it appears mainly on the right side. The weather caster plays an important role in explaining the weather information by using the visual information in the background. Key weather information is sometimes highlighted with captions or colors.

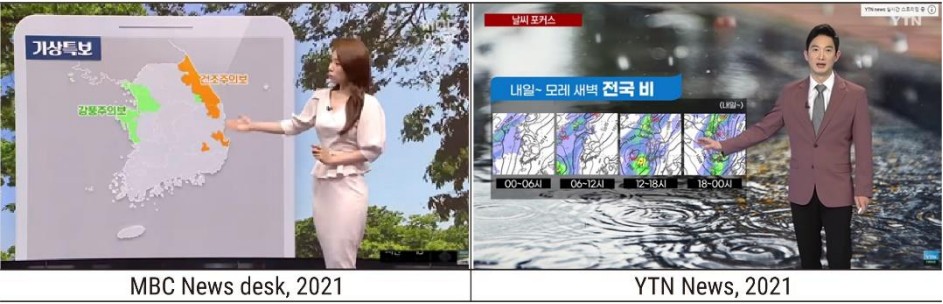

**Figure 7.** Examples of general weather broadcasts without sign language service.

As in the example on the left in Figure 8, a Korean sign language interpreter is placed in the lower right corner, similar to other types of broadcast, or an American sign language interpreter appears in person together with the caster, as in the photo on the right. When an actual sign language interpreter appears together, the weather caster remains on the right and the sign language interpreter is located on the left. Since most of the basic content is arranged for an audience with normal hearing, some of the weather information is usually blocked by the interpreter unless the screen arrangement is changed accordingly.

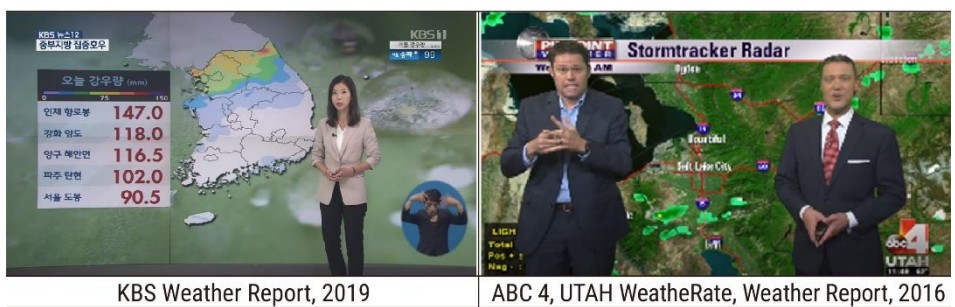

**Figure 8.** Examples of sign language services in weather reports.

*2.2. Elements of Information Delivery and Design of Arrangement Conditions of TV Broadcasting for the DHH People*

By analyzing various TV broadcasting cases that include subtitles or sign language interpreters, information delivery elements for DHH individuals are categorized into (1) sign language interpreter, (2) explanatory subtitles, and (3) reference videos, as shown in Figure 8. As in Table 1, the subordinate options to be considered when arranging these information delivery elements in a rectangular TV screen are designed by category. As for the first element, the sign language interpreter, the layout of the screen (left or right side) and the existence of the background against which the sign language interpreter performs are considered the subordinate options. When multiple speakers appear in a panel discussion, a case in which one interpreter interprets for all speakers and another case in which multiple interpreters are placed in the same screen (one interpreter for one speaker) are considered the options. Regarding subtitles, which is the second element of the information delivery, the presence or absence of subtitles are the subordinate options since most of the TV images usually place subtitles at the bottom center. In the case of multiple speakers appearing, whether each speaker has dedicated subtitles is set as an option as well. As for the reference videos, the third element and the basic reference materials for TV broadcasts, the entire screen and partial screen (with the sign language interpreter blocking some of the reference video screen) are set as the subordinate options.

**Table 1.** Elements of information delivery to be considered in TV broadcasting for DHH individuals.

| Problems with the Placement of Information Transmission Elements for Each Type of Information | | |
| --- | --- | --- |
| **Layout of a sign language interpreter** | | |

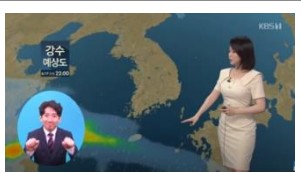 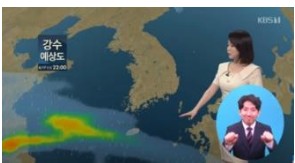

| Left side | | Right side |
| --- | --- | --- |
| **Background of sign language screen** | | |

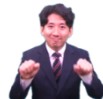 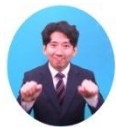 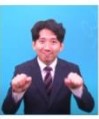

| No background | | Shaped background |
| --- | --- | --- |
| **Number of sign language interpreters** | | |

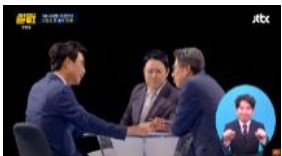 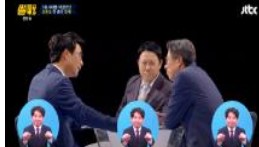 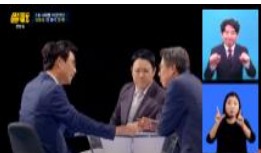

| One interpreter | Multiple interpreter + Horizontal arrangement | Multiple interpreter + Vertical arrangement |
| --- | --- | --- |
| **Number of subtitles according to the number of speakers** | | |

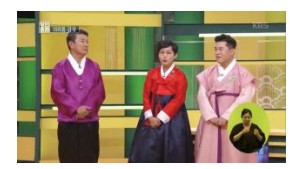 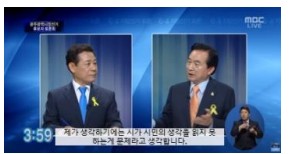 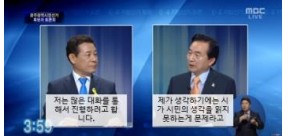

| No subtitle | One subtitle | Multiple subtitles |
| --- | --- | --- |
| **Reference video size** | | |

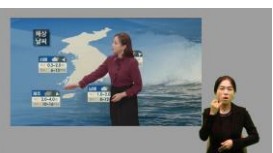 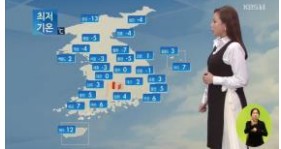

| Partial size | | Full size |
| --- | --- | --- |

Row labels (vertical): Sign language interpreter; Subtitle; Reference video.

## 3. Preferences Evaluation of Information Delivery Elements of TV Broadcasts for DHH Individuals

### 3.1. Elements of Information Delivery and Design of Arrangement Conditions

Experiments for all broadcast types were divided into Test Step 1 and Test Step 2 to conduct the experiment in two stages. In Test Step 1, the arrangement options of visual components were tested by each broadcasts' characteristics, and in Test Step 2, the preference for the presence of subtitles was tested by adding subtitles to each option. For videos fully meeting the conditions for the experiments, a total of 23 experimental videos were created by collecting existing TV clips, filming the sign language interpreter's movements corresponding to the content, and editing the footage as per each of the experimental conditions.

### 3.1.1. Experiment Design for News

For news, the screen was designed focusing on the visibility of the sign language interpreter delivering the content as shown in Table 2. In step 1, the visibility of the interpreter was related to the size and placement of the interpreter. In this type of broadcast, four cases were designed by differentiating the visual components, such as size of the interpreter (small or large size) and background (with or without background), with the most widely used right side layout being set as default. The small size type of the interpreter is the size currently used by TV broadcasts. In step 2, subtitles were added to each of the four cases to examine whether the presence of subtitles was preferred or not.

**Table 2.** Experiment design for news.

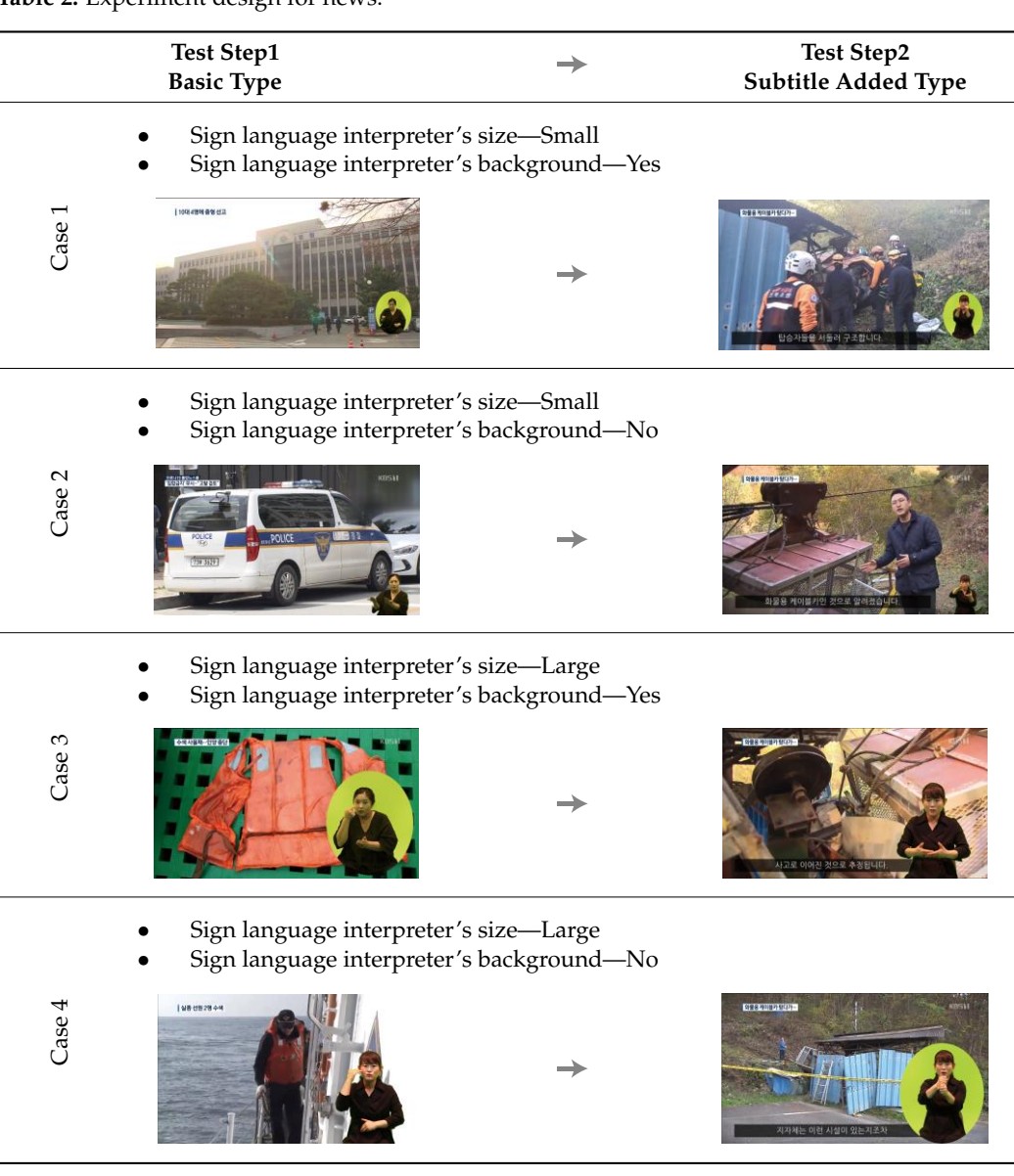

### 3.1.2. Experiment Design for Panel Discussion

The biggest problem for DHH individuals watching panel discussions on TV is that it is difficult to know who is speaking and whose comments are being shown in the subtitles. Therefore, in the panel discussion, the focus was on the speaker-specific sign language and the arrangement of subtitles to allow the audience to easily recognize who's speaking. As seen in Table 3, in step 1, a total of three basic options were provided: (1) one sign language interpreter is provided for each speaker, (2) multiple sign language interpreters

are placed horizontally, and (3) multiple sign language interpreters are placed vertically. After selecting an option in step 1, in step 2, an option of subtitles or individual subtitles for each speaker were also added. In the experiment, video clips of the panel discussion that dealt with politics, society, and entertainment were used without distinction. In the case of a sign language interpreter provided for each speaker, a circular background was used for the horizontal arrangement and a square background was used for the vertical arrangement for efficient use of the screen. In the case of individual subtitles in step 2, the background color and position of the subtitles for each speaker were set different to make it easier to visually distinguish them.

**Table 3.** Experiment design for panel discussions.

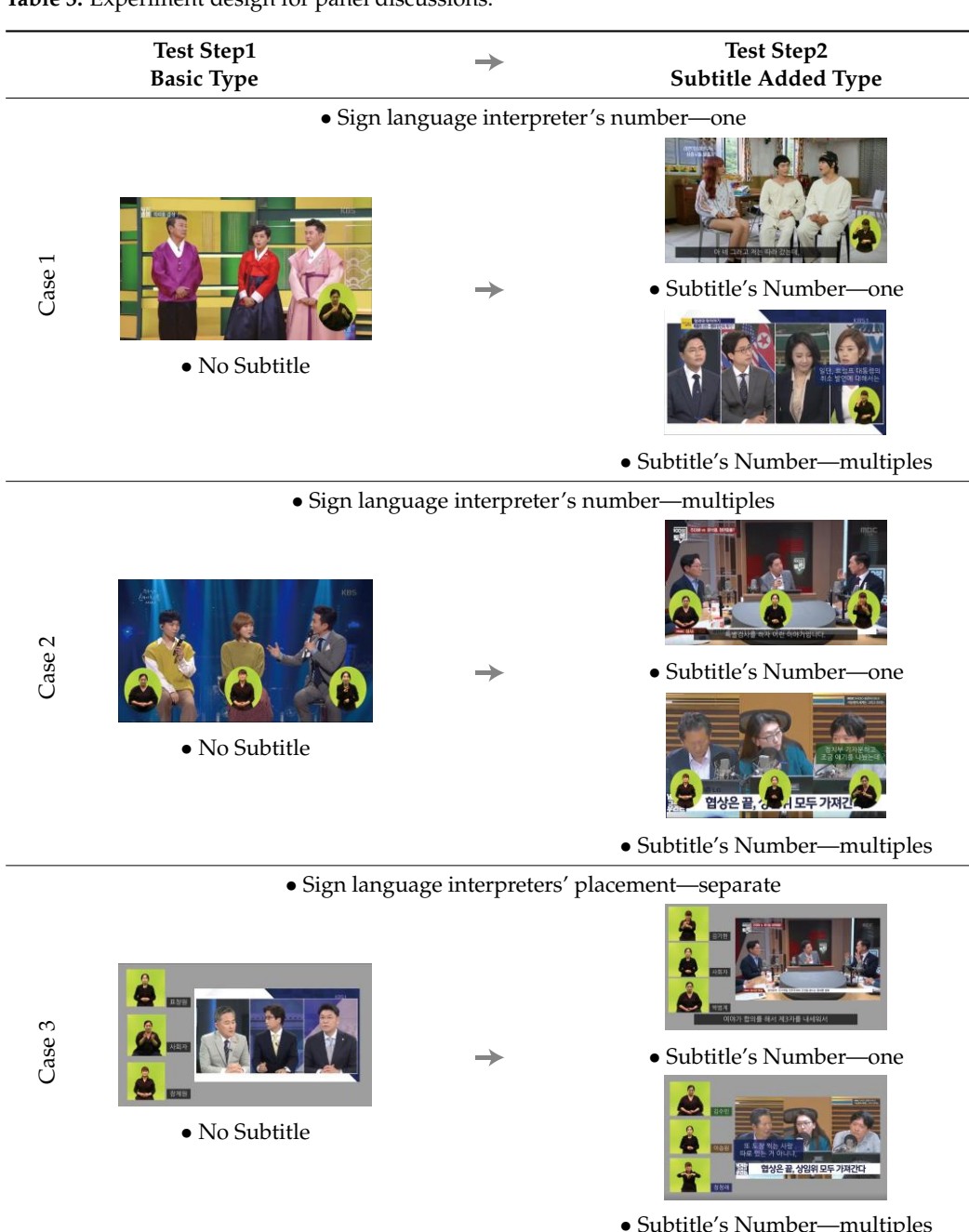

### 3.1.3. Experiment Design for Weather Forecasts

In the weather forecast type, the entire screen consists of weather data and a weather caster who explains the data that appears on one side of the screen, so the placement

(location) of the sign language interpreter is important. With this in mind, in step 1, three basic options were determined: (1) a sign language interpreter is placed on the right side of the weather forecast screen, (2) a sign language interpreter is placed on the left side, and (3) a sign language interpreter and the data screen are presented separately. In step 2, subtitles were added to each option. In the case of a split screen in step 2, both sign language interpreters and subtitles were separated from the weather forecast screen in order to exclude the elements that could interfere with the weather forecast screen. Table 4 presents the examples of the experiment design for the weather forecast type.

**Table 4.** Experiment design for weather forecast types.

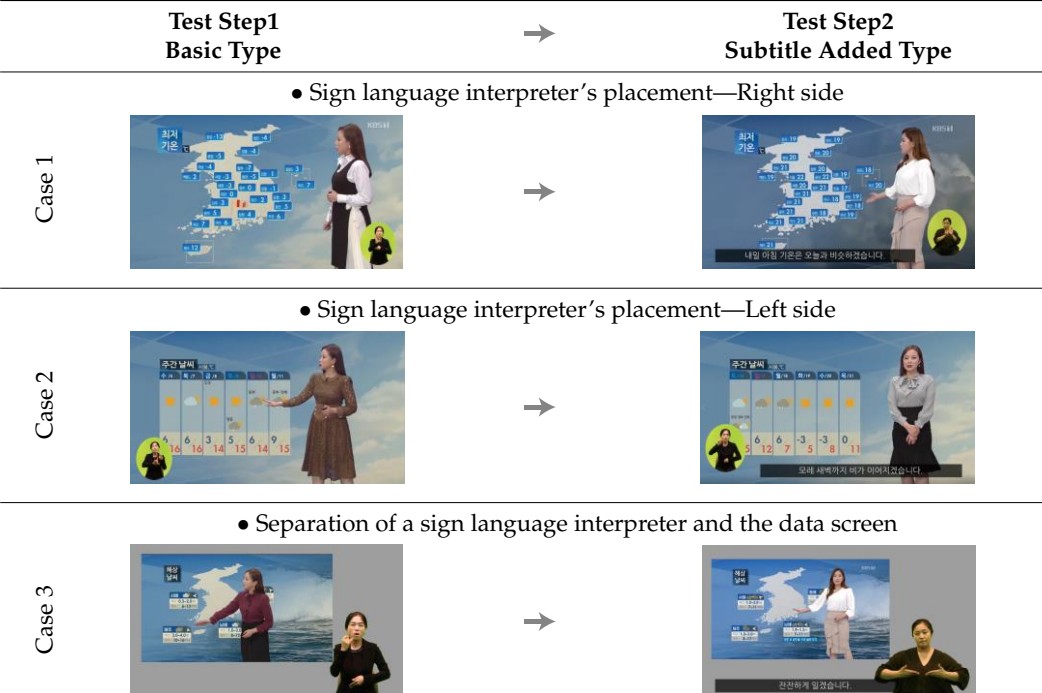

## 3.2. Experiment Environment

This experiment was conducted for 30 DHH participants in two laboratories as shown in Figure 9 at the shelter for the DHH people run by the Gwangju city branch of the Korean Association of the Deaf. The subjects of the experiment were people with hearing impairment over the age of 20; the age range of subjects was from 30 to 60 and there were 24 females and 6 males. Regarding educational background, there were those without education (2), middle school graduates or lower (8), high school graduates (11), and college graduates (2). The experiment consisted of a total of three people: a subject, a sign language interpreter to help guide the experiment progress, and an investigator. Tobii Nano, an eye tracker, was attached to a 27-inch monitor to collect gaze information from the subject during the experiment.

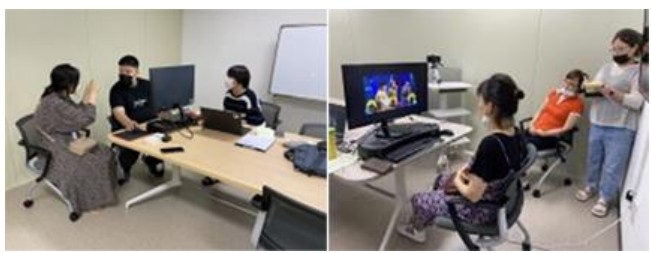

**Figure 9.** Test environment.

Prior to the experiment, a survey was conducted on the level of the use of sign language and Hangeul (Korean language) among the subjects. The results showed that 93.5% of the 30 subjects had used sign language for more than 10 years, but only 74.2% of them had received regular sign language education before. A total of 45.2% of the subjects answered that there was an expression they did not know when using sign language. In addition, when asked about the use of character language, 71% of them answered that they had received Hangeul education and used Hangeul for more than 10 years. However, 83.9% answered that they felt difficulties due to the different systems of Hangeul and Korean sign language.

In this experiment, the subject was asked to select the most preferred subordinate option after watching the step 1 video clips for each broadcast type as described above. In step 2, they watched a video with subtitles added to the layout type selected in step 1. After completing step 2, they were asked to evaluate their preferences for all layout types that they watched in the 2 steps of the experiments. In addition, individual interviews were conducted to investigate the reasons for their preferred layout type. Each subject was asked to evaluate two types at random. An average of 20 subjects responded per type. For the eye tracking of the subjects, the eye movement path of the subjects and the frequency of gaze for each component of the screen were identified by using the screen-attached eye tracker "Tobii nano", which is a convenient device for tracking the user's gaze on the monitor screen [24].

Initially, the experiment was designed with each subject answering a total of three phases of questions: one for each type (news, panel discussion, weather forecast). However, this took more than one and a half hours since the majority of the subjects were found to experience difficulties understanding and answering the questions after watching the video clips. A sign language interpreter helped them understand the experiments, but the experiment did not progress as smoothly as expected. Considering the fatigue of the subjects, the experiment design was changed so that one subject watched two clips for each type, and an average of 20 subjects participated in watching and answering sessions for each video type. Compared to the subjects with normal hearing, the experiments on the subjects with hearing loss took two to three times longer.

## 4. Experimental Results

The questions presented to the subjects for the evaluation of preference for various layout components are shown in the Table 5 below. Among these, question 5 is to select the screen that was most uncomfortable to see, and, unlike questions 1 to 4, it is a negative question.

**Table 5.** Questions for preference evaluation of the screen layout components.

| No | Key Consideration | Question |
|----|-------------------|----------|
| 1 | Efficiency in arrangement of information delivery elements | On which screen were the information delivery elements arranged most efficiently? |
| 2 | Harmony in arrangement of information delivery elements | On which screen were the information delivery elements arranged most properly and harmoniously? |
| 3 | Visibility of the sign language interpreter | On which screen was the interpreter the most visible? |
| 4 | User preferences | Which screen do you want to continue to use? |
| 5 | Message delivery | Which screen was the most difficult to understand the content of the message? * (* Negative question) |

Prior to the comparative analysis for each type, we evaluated the inter-scorer reliability of the response's scores for each question. Acceptable Cronbach's alpha for three types (news, panel discussion, and weather forecasts) were obtained as 0.76, 0.68, and 0.77, respectively.

### 4.1. Results of Step 1 by Type

4.1.1. News

The results of the survey conducted after the subjects watched the video clips containing accidents (news) are as follows.

For the first question ("On which screen were the information delivery elements arranged most efficiently?"), as shown in Figure 10, 14 subjects selected clip no. 4, 5 subjects selected clip no. 3, and 1 subject each selected clip no. 1 and no. 2. For the second question, ("On which screen were the information delivery elements arranged most properly and harmoniously?"), 15 subjects selected clip no. 4 and 5 selected clip no. 3. For question no. three, ("On what screen is the sign language interpreter most clearly visible?"), 13 subjects selected clip no. 4, 7 subjects selected clip no. 3, and no one selected clip no. 1. This suggests that the majority preferred to have a large sign language interpreter on screen without a background.

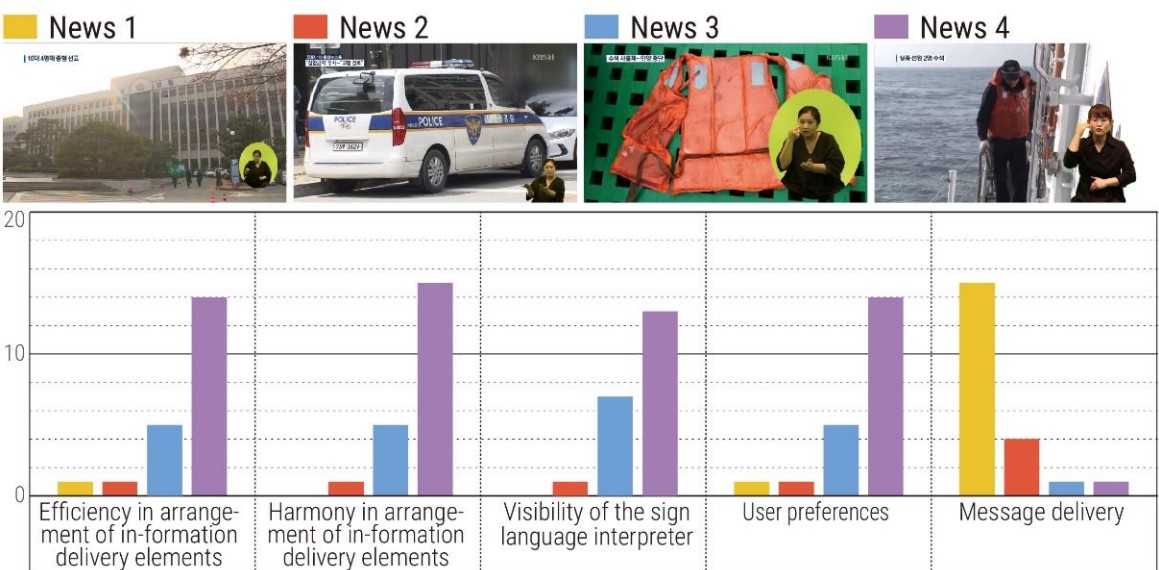

**Figure 10.** Results with news cases.

For the fourth question ("Which screen do you want to continue to use?"), 14 subjects selected clip no. 3, 5 subjects selected clip no. 4, and 1 subject each selected clip no. 1 and no. 2. For the fifth question, ("Which screen was the most difficult to understand the content?"), the overwhelming majority of the subjects chose clip no. 1, which presented the layout with a background and a sign language interpreter in small size. Considering that the majority of subjects selected clip no. 1 and no. 2, it was found that the size of the sign language interpreter was a more important consideration than the presence of a background.

To summarize, as shown in Figure 11, the average value of news type preferences showed that the best screen layout for the news type was clip no. 4 with a large-sized sign language interpreter and no background of the interpreter, and the worst screen was clip no. 1, which is similar to the one currently shown on TV programs today.

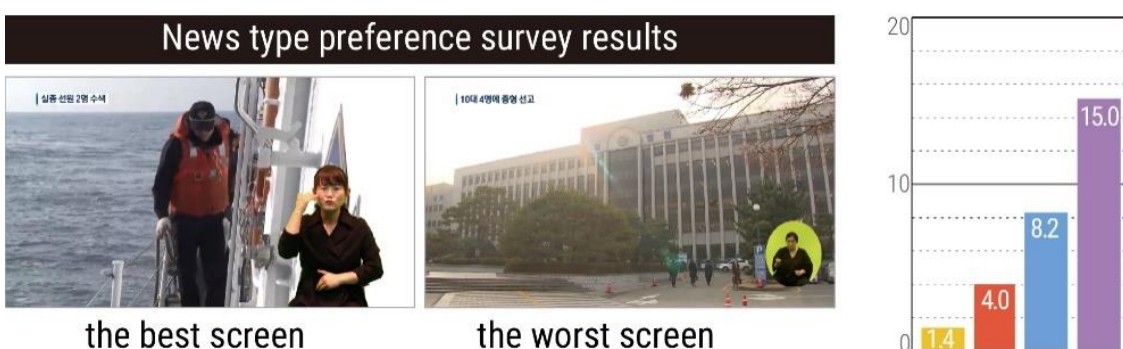

**Figure 11.** Results of preferences evaluation with news cases.

In an individual interview after the preferences survey, as presented in Table 6, the subjects presented the following opinions on the screen components of news type and the proposed screen layout.

**Table 6.** Subjects' opinions on the news clips.

| | |
|---|---|
| **Size of the Interpreter** | The size of the sign language interpreter for clip no. 1 (news with an accident piece), which was similar to the current sign language interpretation service, was too small to see. Clip no. 3 and no. 4, however, had a larger size of interpreter, making it easier to recognize the sign language. |
| **Presence of Background** | There was no background behind the sign language interpreter in clip no. 2 and no. 4, making it cleaner and much easier to see because it does not give this feeling of being trapped in a space. As in news clips no. 1 and no. 3, there always is a background to the sign language interpreter, so I felt kind of stuffy because the sign language interpreter seemed to be confined to a small space. |
| **Other** | News clip no. 1, which is similar to existing sign language services, had a small-sized sign language interpreter and was difficult to see along with the video footage because the interpreter was placed in a corner. The blue chroma key background, which is mainly used as the background color, was so intense that it was difficult for my eyes to see the sign language against it. (There were opinions that the colors used in the experiment were too intense as well, which made their eyes tired.) |

Figure 12 shows the result of the gaze-tracking hit map in news cases. In the subtitled and most preferred layout, gaze was mostly focused on the subtitles and sign language interpreter. The subjects often identified professional words from the subtitle since sign language is limited in the sufficient delivery of the information. In the interview, a subject mentioned that it is essential to read subtitles because sign language does not contain many professional words used in news broadcasts. On the other hand, subjects least preferred watching the main content and the subtitled layout rather than watching the subtitles. If the presentation of the interpreter is too small, the subjects gave up trying to figure out the full content and, thus, watched fewer subtitles.

### 4.1.2. Panel Discussion

As shown in Figure 13, in the panel discussion type, 12 subjects selected clip no. 1 and 8 subjects selected clip no. 2 for the first question ("On which screen were the information delivery elements arranged most efficiently?"). For the second question, ("On which screen were the information delivery elements arranged most properly and harmoniously?"), 11 subjects selected clip no. 2 and 8 selected clip no. 1. For the third question, ("On which screen is the sign language interpreter most clearly visible?"), 10 subjects each selected clip no. 1 and no. 2, respectively.

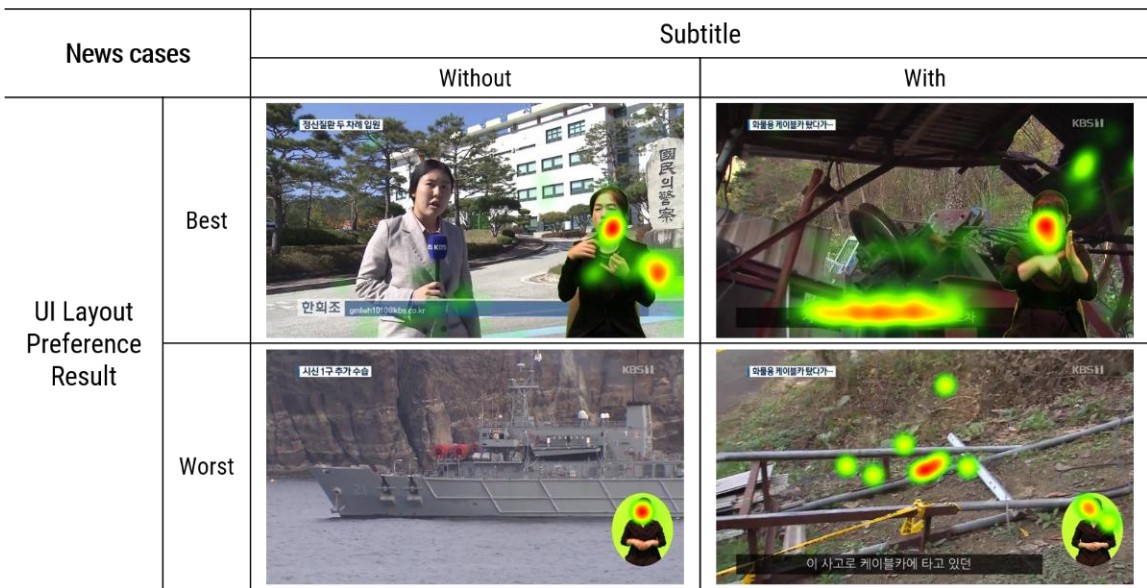

**Figure 12.** Results of eye tracker experiments with news cases.

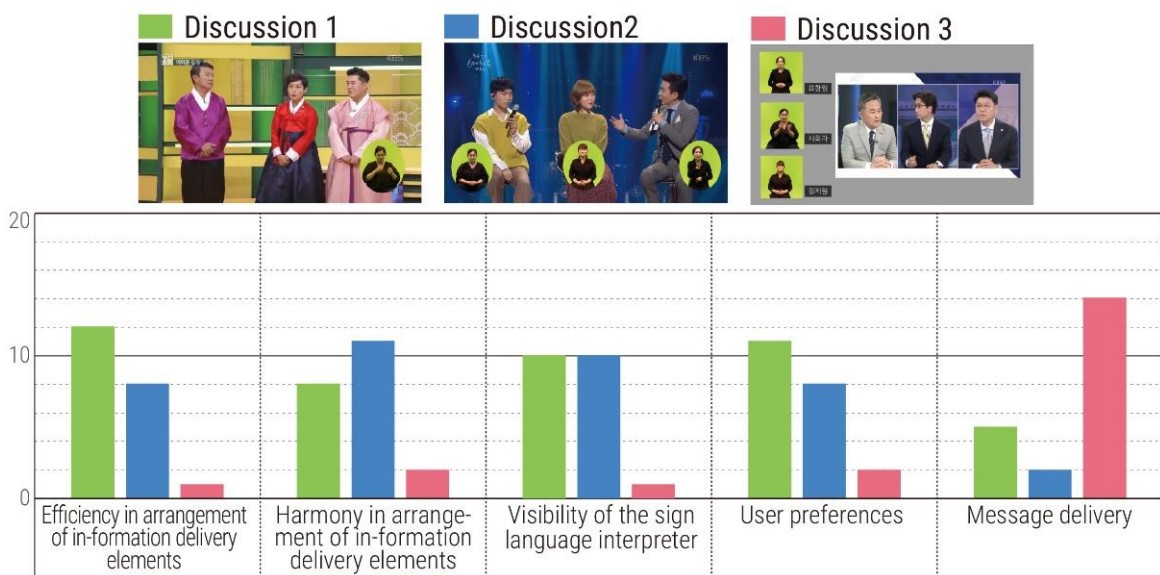

**Figure 13.** Results of experiments in panel discussion cases.

For the fourth question ("Which screen do you want to continue to use?"), 11 subjects selected clip no. 1, and 8 subjects selected clip no. 2. For the fifth question, ("Which screen was the most difficult to understand the content?"), clip no. 3 was selected the most, where the sign language screens were provided for each speaker and arranged vertically. A total of 14 subjects chose clip no. 3, and 5 selected clip no. 5.

When looking at the average value of preference in the panel discussion type, as shown in Figure 14, basic type 1 and type 2 showed a similar level of preferences with 11.2 and 11.0, respectively. In an interview after the experiment, some subjects commented that there is a shortage of sign language interpreters in reality, and it is wasteful and not economical to have multiple interpreters on the same screen all at once. Clip no. 3, in which multiple sign language interpreters were vertically arranged separately, got a low score because it was difficult and confusing to match the speaker and the interpreter.

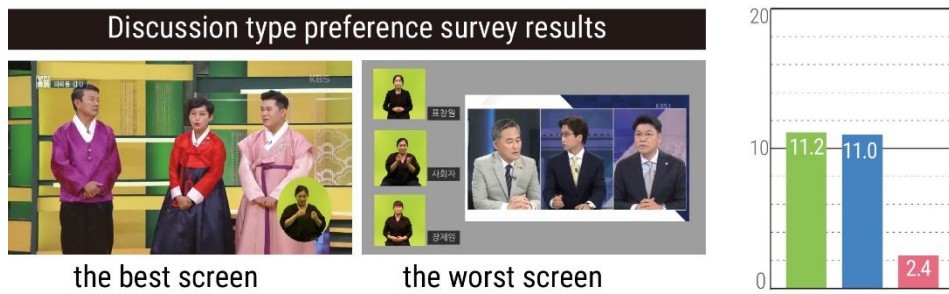

**Figure 14.** Results of preference evaluation with discussion cases.

In an individual interview after the preferences survey, as presented in Table 7, the subjects presented the following opinions on the screen components of panel discussion cases and the proposed screen layout.

**Table 7.** Subjects' opinions on the panel discussion clips.

| | |
|---|---|
| **Number of Interpreters** | The conventional method shown in discussion screen clip no. 1 is familiar. However, it is difficult to see who is speaking with only one sign language interpreter performing. |
| **Horizontal Arrangement** | The arrangement of sign language interpreters for each speaker on clip no. 2 seems to be wasteful and not economical since the supply of sign language interpreters is in shortage. Yet, it helps me better understand who is speaking. |
| **Vertical Arrangement** | For discussion clip no. 3, where multiple sign language screens were arranged vertically, there was an opinion that it was too complicated to see while going back and forth between the sign language screen and the original screen, and that it was also difficult to match the sign language interpreter and the speaker. |

Figure 15 shows examples of the gaze-tracking hit map for the panel discussion case. In the most preferred layouts, subjects fairly focused on the interpreter, however, in the worst layouts, the gaze was scattered and there was much transition between speakers and the corresponding interpreters to figure out who is speaking.

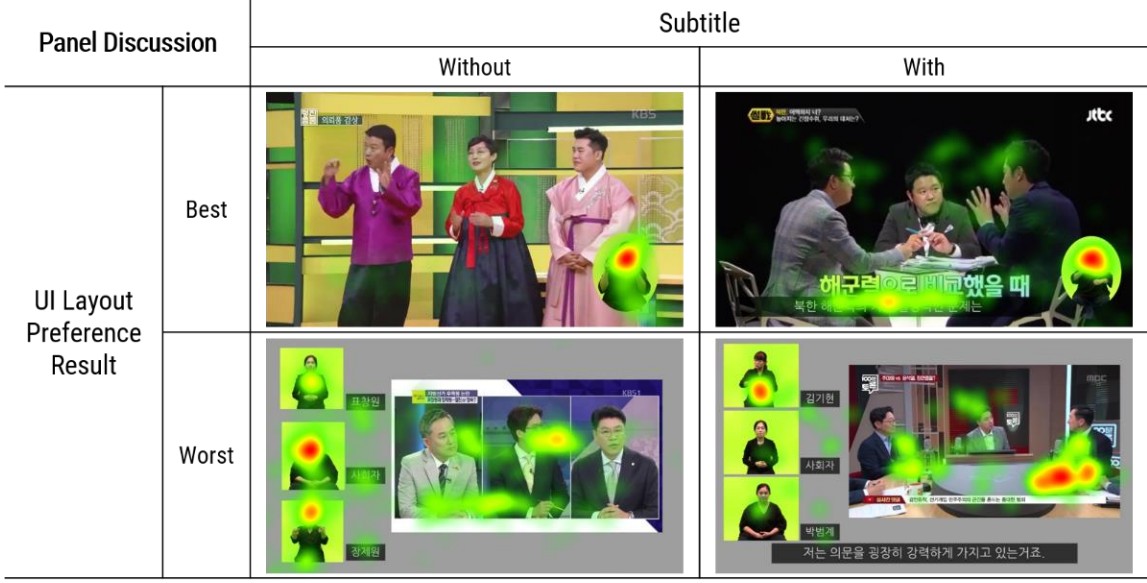

**Figure 15.** Results of eye tracker experiments with discussion cases.

4.1.3. Weather Forecasts

For the first question ("On which screen were the information delivery elements arranged most efficiently?"), as shown in Figure 16, 14 subjects selected clip no. 3, 5 subjects selected clip no. 1, and 3 subject selected clip no. 2. For the second question, ("On which screen were the information delivery elements arranged most properly and harmoniously?") and the third question ("On which screen is the sign language interpreter most clearly visible"), 12 subjects selected clip no. 3 and 5 each selected clip no. 1 and no. 2. For the fourth question ("Which screen do you want to continue to use?"), 13 subjects selected clip no. 3, 5 chose clip no. 1, and 4 selected clip no. 2. For the fifth question, ("Which screen was the most difficult to understand the content?"), 8 subjects each chose clip no. 1 and no. 2 in which the interpreter was arranged differently. Clip no. 3, which is a split arrangement of the data screen and sign language interpreter, was selected by 6 subjects, showing that all three types of clips were preferred by a similar number of the subjects.

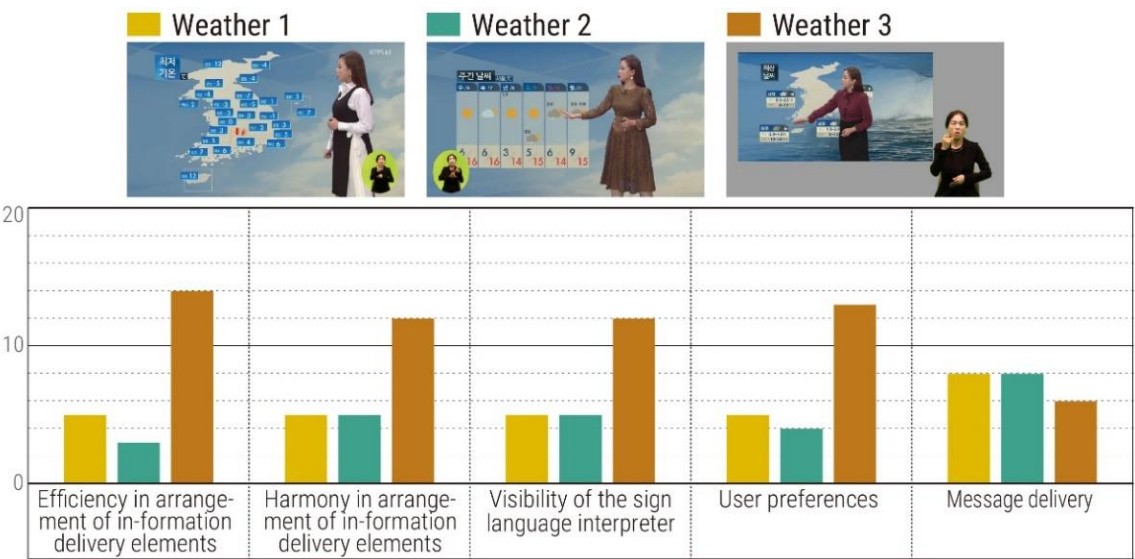

**Figure 16.** Results of experiments in weather forecast cases.

As for the average value of the preference of the weather forecast type, as shown in Figure 17, clip no. 3, with the weather description part split from the large-sized sign language interpreter, showed the highest preference, and clip no. 2, with the sign language screen being placed on the opposite side of the weather caster, was at the lowest in preference. In the case of clip no. 1, which has a slightly higher score than clip no. 2, the subjects' opinions were that the lower right arrangement of the sign language interpreter (like the current broadcast sign language services) is familiar, and it is easy to see as the interpreter was in the same direction as the weather caster.

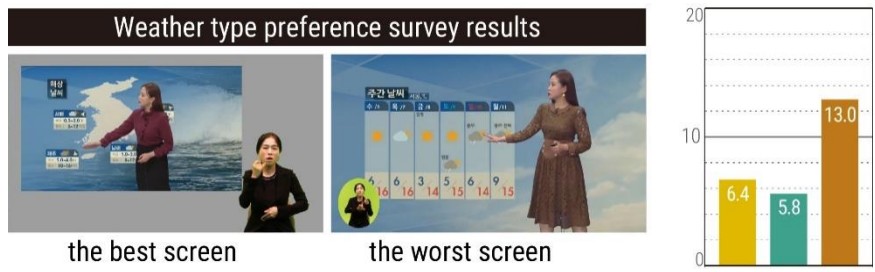

**Figure 17.** Results of preference evaluation with weather report clips.

They answered that it would be difficult to see if the weather data were too small because the entire screen is a visual reference containing information. They also answered

that the visibility of the weather data was more important than the sign language interpreter or subtitles for weather report broadcasts. In addition, they gave the opinions that the sign language screen should not cover the hands of the weather caster because a weather caster's hands indicating the weather data serves an important role in weather forecasts.

Figure 18 shows the result of the gaze-tracking hit map in the weather forecast case. The subjects mostly focused on the interpreter and sometimes watched subtitles or the weather information. Compared to the other two cases, the dependency on the subtitle is small, since the presentation and content of the weather information is very formal and easy.

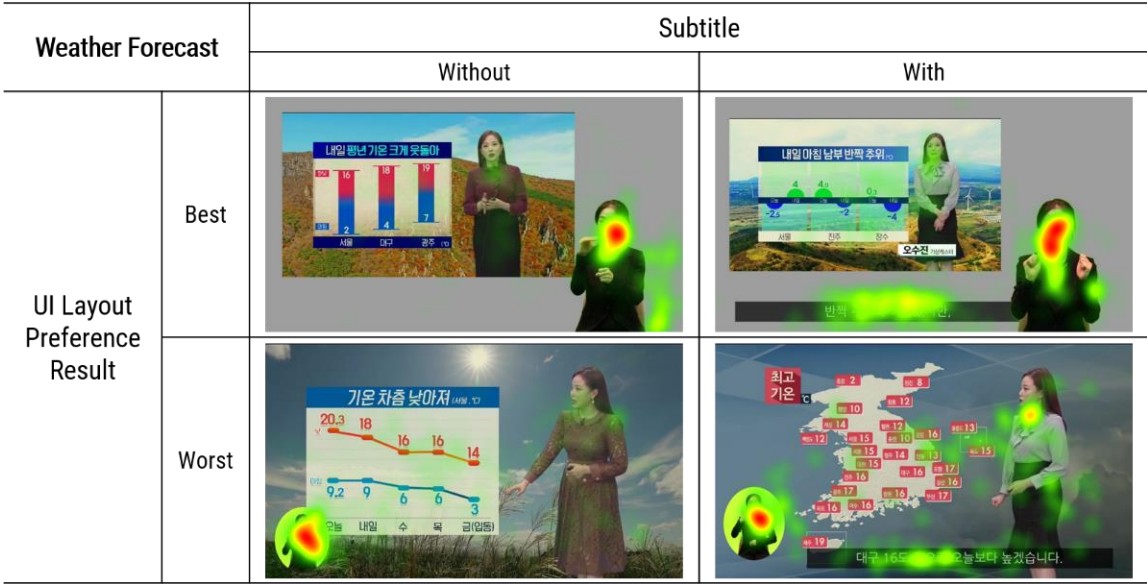

**Figure 18.** Results of eye tracker experiments with weather report cases.

In individual interviews, the subjects presented the following opinions on the screen components of weather report clips and the proposed screen layout as presented in Table 8.

**Table 8.** Subjects' opinions on the weather report clips.

| | |
|---|---|
| **Sign Language Screen Arrangement** | For clip no. 2, it's awkward because I'm not used to the location of the sign language interpreter there. The sign language interpreter and the weather caster are positioned too far, so it's very inconvenient to see. It was good though that the sign language screen has been moved so it does not block the contents of the screen. |
| **Size of the Reference Footage** | Clip no. 3 has a smaller data screen, but the size of the interpreter is larger, making it easier to understand and more convenient. |

### 4.2. Results of Experiments with Subtitles

In response to the question, "Choose the one with or without subtitles that you think is useful, and why?", the subjects answered that they preferred the presence of subtitles for all three types of clips, as shown in Figure 19.

Regarding subtitles, as presented in Table 9, the subjects responded that subtitles and sign language should be provided together to help people with hearing impairment acquire information, and that subtitles also help them to learn unknown words or vocabulary. Some mentioned the problem of the arrangement and speed of subtitles, saying that the speed of subtitles was too fast for DHH individuals and hard to see at the same time with sign language. Therefore, many commented that it would be better to provide 2–3 lines of subtitles instead of one line, allowing the subtitles to stay on the screen longer, so that people do not miss the content while watching it alternately with the sign language.

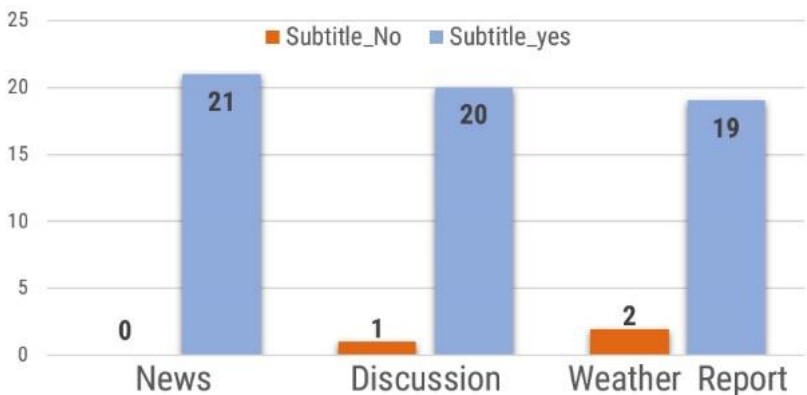

**Figure 19.** Results of experiments with subtitles.

**Table 9.** Subjects' opinions on the subtitles.

| | |
|---|---|
| **Role of the Subtitles** | It's better when subtitles and sign language are provided together. With subtitles, you can study vocabulary or sentences that you don't understand. You can see both sign language and subtitles to get a more accurate understanding of the meaning. |
| **Arrangement of the Subtitles** | If the distance between the sign language interpreter and the subtitles is far, it is inconvenient to see both since your eyes are moving back and forth. Since the subtitles pass by too fast, it would be better to have 2–3 lines rather than 1 line on the screen. |

Regarding the experiments with subtitles added to the three types of video clips, all subjects answered that the subtitles are necessary for the news clips. They responded that there may be unfamiliar expressions if only a sign language interpreter is provided, but when subtitles and sign language interpretation are provided together, this gives them an opportunity to understand or learn difficult words. Looking at the eye tracker data analysis results for news type footage, some subjects looked at the subtitles more than the sign language interpreters.

For the panel discussion type, three conditions were tested: (1) No subtitles, (2) One representative subtitle, and (3) Individual subtitles for each of the multiple speakers. As shown in Figure 20, 2 out of 21 subjects answered that it was easier to see the clips without subtitles, and 19 answered that they liked it better when subtitles are provided. Among the 19 subjects who answered they preferred the presence of the subtitles, 6 said they preferred one integrated subtitle and 13 preferred individual subtitles for each speaker.

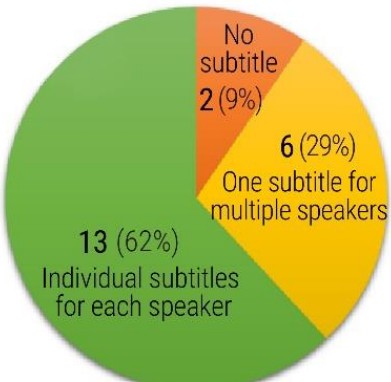

**Figure 20.** Preference on integrated or individual subtitles for multiple speakers.

Many answered that arranging multiple sign language interpreters for each speaker, as proposed in the basic type in step 1 for the panel discussion clip, was confusing and financially not feasible. The subjects' opinion was that it would be very good to have one sign language interpreter and individual subtitles for each speaker. A case where there is only one sign language interpreter performing while placing multiple subtitles separately for each speaker in the panel discussion, as shown in Figure 21, showed the highest preference at 62%. Most of the subjects responded that it was less confusing and easy to understand which speaker was talking when subtitles appeared for each speaker. There was one opinion that it would be nice to provide subtitles in the form of speech bubbles.

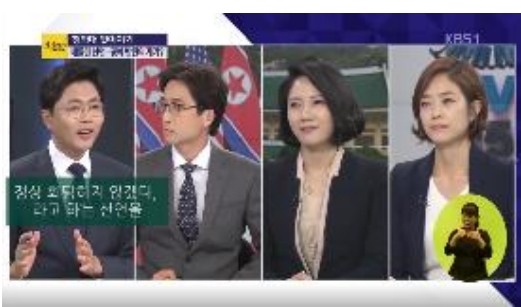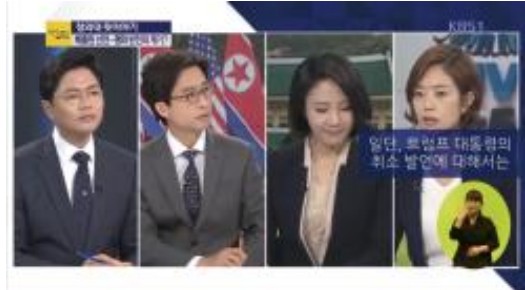

**Figure 21.** Example of individual subtitles for each speaker.

In the weather forecast type, most of the subjects preferred the presence of subtitles, while 9% of the respondents preferred the clip with no subtitles. They said the reason is that subtitles are not necessary because the sign language interpretation and the reference weather report data provided can be enough, to some extent, when it comes to weather reports.

In conclusion, based on the results of experiments and subjects' opinions, the role or necessity of subtitles can be identified. Therefore, subtitles are necessary and should be provided in 2–3 lines and stay longer on the screen so that DHH individuals can read them along with a sign language interpretation. In panel discussions with multiple speakers appearing, the need for individual subtitles was also found.

## 5. Design Proposal of TV Broadcasting for DHH Individuals

With the eye tracker data collected on the screen navigation from the three types of TV broadcasting, it was found that DHH participants viewed the sign language interpreter more than the reference video or subtitles in all cases. In addition, DHH participants tended to look at the face of the sign language interpreter the most, because not only the interpreter's hand shapes but also the facial expressions of the interpreter are very important when comprehending the message. This is because, even for the same hand shapes, the viewer can acquire additional information (i.e., whether the content is positive or negative) depending on the interpreter's facial expression. As shown in Figure 22, the sign language interpreter is currently broadcast in about a quarter of the size of the screen. This is not the appropriate size for viewers to see facial expressions. Therefore, the sign language interpreter should be placed as large as possible on the screen. Moreover, since DHH participants do not prefer the background behind the interpreter, the interpreter would be better placed without a background.

Based on the results from the experiments conducted above, the basic conditions for the information delivery component layout design for DHH individuals are presented in Table 10.

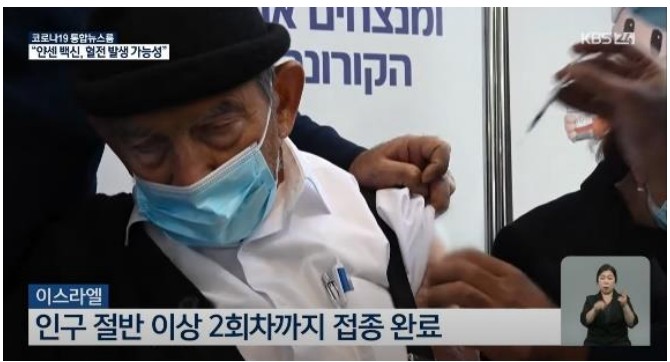

**Figure 22.** Example of the current sign language interpreter size in TV broadcasts.

**Table 10.** Basic conditions for information delivery component layout design for DHH individuals.

| Category | | News | Panel Discussion | Weather Forecasts |
|---|---|---|---|---|
| Sign Language interpreter | Size | At least 1/3 of the screen's height (Larger than as-is size) | At least 1/3 of the screen's height (Larger than as-is size) | At least 1/3 of the screen's height (Larger than as-is size) |
| | Background | Not necessary | Not necessary | Not necessary |
| | Location | Right side | Right side | Right side |
| | No. required | 1 interpreter | 1 interpreter | 1 interpreter |
| Reference video | | No need to split the screen | No need to split the screen | Split screen preferred |
| Subtitles | | Necessary | Necessary (Individual subtitles preferred) | Necessary |

Based on the above results, the layout design for each information type for DHH individuals was organized.

In the case of the news type, the opinion of most of the subjects that the size of the sign language interpreter was too small was respected. For news type clips, the sign language interpreter is the most important component in information delivery because DHH individuals acquire information through sign language rather than the reference footages provided. As shown in Figure 23, the interpreter size is enlarged to 1/3 the height of the full screen so that sign language can be seen clearly, improving visibility compared to existing ones. The background of the interpreter has been removed so that the reference footage is not unnecessarily blocked, and the viewers can feel less stuffy watching the screen. For the subtitles, two lines are provided at a time allowing the subtitles not to pass too quickly so that DHH individuals can look at both sign language and subtitles together.

For panel discussions with multiple speakers, the subjects' opinion that multiple interpreters appearing can be confusing and impractical was reflected in the proposed layout. As seen in Figure 24, only one interpreter is placed on the screen. Subtitles are provided for each speaker so the viewers can clearly understand each speaker's comment. In the layout, it is important that the subtitles are not blocking the mouth or hands of the speakers. It is easy to identify the speaker when the subtitles are color-coded by speaker, so translucent subtitle boxes of various colors have been used.

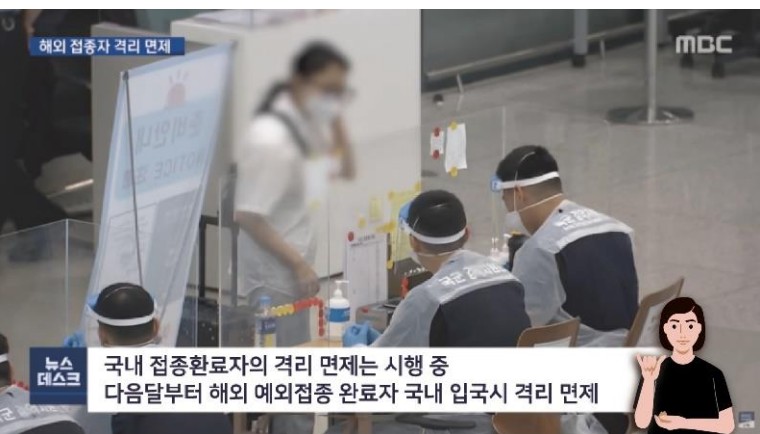

**Figure 23.** Proposed layout design of information delivery components for news type broadcasts.

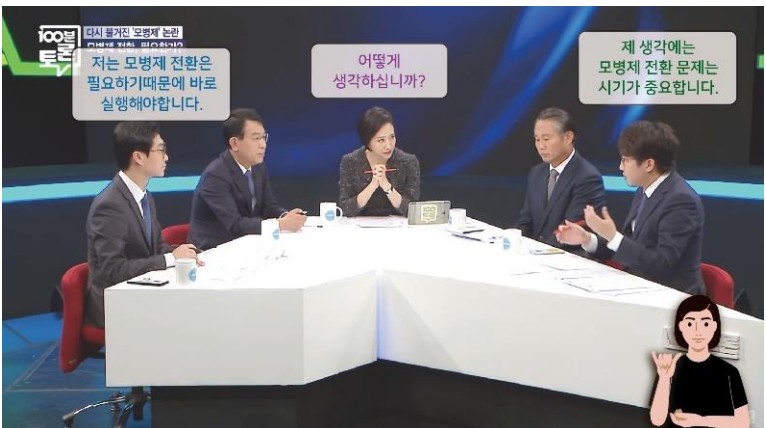

**Figure 24.** Proposed layout design of information delivery components for discussion type broadcasts.

For weather forecast types where the weather data and caster are presented in the same screen, it is recommended to reduce the size of the weather data section and make it separate from the sign language interpreter, as shown in Figure 25, so that the interpreter does not block the weather data section. A blank space is created on the screen because of this. In this blank space, a sign language interpreter and subtitles are placed. Both the weather caster and sign language interpreter are placed on the right side on the screen so that the movement of the viewers' eyes can be minimized when watching the weather forecast. The sign language interpreter should be careful not to block the weather caster and his/her hands or indications. Although some answered that subtitles are not highly necessary for weather forecast types, the overall survey results showed that most of the respondents think that subtitles are required. Therefore, two to three lines of subtitles are placed on the screen that do not block other information on the screen.

One general cautionary note for all broadcast types is that improving the visibility of sign language interpreters should be the priority, and subtitles are considered next. Although 29% of the subjects answered in the pre-interview that it is important and necessary that sign language and subtitles appear together, the results of eye tracker analysis showed that the time the subjects stared at the subtitles was shorter than the time they spent looking at the sign language interpreter. From this, it can be assumed that sign language acquired as the native language is the main tool for DHH individuals to acquire key information. Therefore, it is necessary to increase the visibility of sign language interpreters and provide subtitles to expand information acquisition opportunities for DHH people who have acquired sign language as their native language.

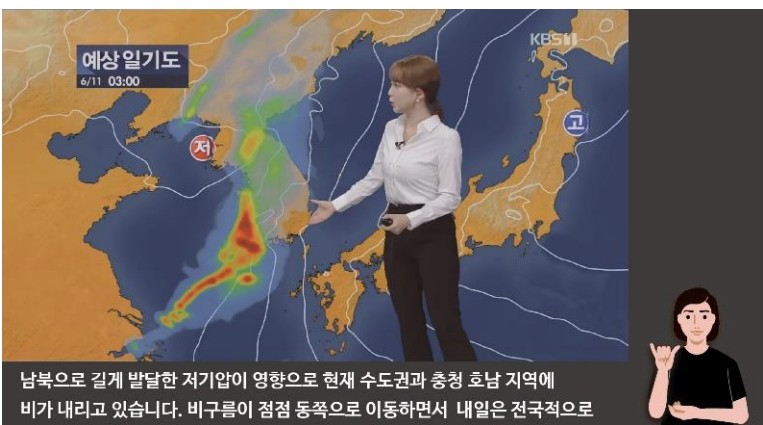

**Figure 25.** Proposed layout design of information delivery components for weather forecasts.

## 6. Conclusions

This study was conducted to analyze and find solutions to the difficulties experienced by DHH people in viewing public TV broadcasts alongside non-disabled people. The process of discovering and resolving these problems was the starting point for understanding how DHH individuals communicate information. Through research and experimentation, three important considerations have been identified from this study. The first is the importance of subtitles. Contrary to the assumptions of non-disabled people that DHH people do not see subtitles but resort only to sign language, they are watching subtitles and sign language together, and the subtitles can also serve as an opportunity for DHH people to learn. This is a problem that DHH people experience because the language system of sign language is different from that of Hangeul. This also indicates that providing subtitles can serve as an opportunity for DHH people to understand the language system of non-disabled people.

Second, sign language can be understood accurately when a viewer looks at the hand shapes and facial expressions of the performer together due to the linguistic limitations of sign language, which conveys meaning only by hands. Third, DHH participants find the background of the sign language interpreter uncomfortable. The investigator (and as a designer) assumed that distinguishing the sign language interpreter by using a background would increase visibility from the screen, but DHH participants felt that the interpreter is confined to a small space when there is a background. Some also responded that the background color made their eyes tired, thereby interfering with watching sign language. Contrary to the general conception that there would be no problem for DHH people in viewing TV broadcasts simply by placing a sign language interpreter on the screen, the results of this study found that a more delicate screen layout design is necessary for DHH people.

It is expected that this study will serve as a helpful guide in providing better sign language services for TV broadcasts that can be conveniently viewed by both DHH people and non-disabled people. However, the experiments were conducted only on DHH people in this study. Therefore, in the future study, it is necessary to analyze the preference and satisfaction of non-disabled people who watch sign language services for TV broadcasts as well. To that end, the scope of this study should be further expanded to find ideal TV news layout designs that can be enjoyed both by DHH people and non-disabled people. Additionally, many other factors, e.g., background with illumination variations or variations of the sign interpreter in age and sex, could be investigated as a part of the future work to find a better design for TV broadcasts.

**Author Contributions:** Conceptualization, J.H.Y. and J.-H.H.; Formal analysis, Y.-G.N. and S.O.; Methodology, J.H.Y. and S.K.; Software, Y.-G.N.; Supervision, J.-H.H.; Validation, S.K. and S.O.; Writing—original draft, J.H.Y. and S.K.; Writing—review & editing, J.H.Y. and J.-H.H. All authors have read and agreed to the published version of the manuscript.

**Funding:** This research was supported the National Research Foundation of Korea (NRF) funded by the MSIT (2021R1A4A1030075).

**Institutional Review Board Statement:** Not applicable.

**Informed Consent Statement:** Not applicable.

**Conflicts of Interest:** The authors declare no conflict of interest.

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
