# Peer review of "Design Proposal for Sign Language Services in TV Broadcasting from the Perspective of People Who Are Deaf or Hard of Hearing"

_applsci, doi:10.3390/app112311211_

Round 1
Reviewer 1 Report
An interesting study!
The study, however, lacks valid and systematic statistical analysis to support the conclusions made by the authors. Presentation and analysis in terms of the statistics in the paper should be upgraded to the level generally required by science journals.
The conclusions and proposals need to have valid support from analysis, with a statistical analysis of variance showing any significance in differences between the arrangements, for example. The identification of the best and worst screen arrangement with a sign language interpreter shot does not mean anything if they are not significantly different in evaluation values in any of the information delivery types.
Presentation of sign language interpretation and subtitles would provide information necessary for the people with hearing loss and is indispensable, but the arrangement also would affect the viewers without disabilities. The study should have surveyed the people without disabilities to accept the proposed arrangement. For example, the proposed layout design for information delivery in Figure 24 would not be accepted and supported by the broadcasting companies nor by the viewers without disabilities or with other type of disabilities.
Author Response
As the reviewer commented, it would be good to have a future work to study the proper layout for both DHH and non-disabled people. Recently, TV programs like news have been providing sign language services, thus we have focused on investigating how DHH people think about the layout of sign language services added to the news screens from the perspective of DHH people. We have pointed out the point in the Conclusion.
We agree the lack of statistical analysis as the reviewer mentioned. Despite practical difficulties carrying out a user study with DHH people, we believe our results and proposals based on the survey results, interviews and gaze analysis provide insights to design a better layout with sign language services for DHH people.
Reviewer 2 Report
Overall, this is an important study with a good design! It is essential to consider the perspectives of the Deaf community themselves in the establishment of appropriate services. To that end, the most obvious concern of the paper is cultural misconceptions and inappropriate terminology, despite the overarching aims to be culturally inclusive.
For example, use of “hearing impairment” (lines 38, 76, 257….)
“communication problem” (line 67…)
“hand gestures” (line 479….)
The description that hearing loss impacts language learning and the assumption that this is a direct cause of low literacy levels is presumptuous. (Beginning on line 66.) In fact, it is the lack of access to language (spoken or signed) rather than the hearing loss itself that begins this trajectory. (However, the description that begins around line 80 is much more appropriate!)
Along those lines, this paper would benefit significantly from clarity within the introduction as to the geographically-specific Deaf community that it targets and actual sign language that was used. (Korean Sign Language?) As a reader, it left me wondering until almost the end of the paper (and even still unsure) about where this study took place. To that end, if there are cultural differences between the Korean Deaf Community/Language and that of the United States Deaf Community (such as acceptance of terminology), those should be clearly acknowledged. However, if the original publication is to be in English, I would strongly advise to use the culturally appropriate terminology.
I am not sure that “easily” is the best term to describe accessibility of TV broadcasting to “everyone.” I would recommend replacement with “typically.” (Line 42.)
Some reference-heavy claims are missing citations, such as lines 47-49; 52-65.
I think that the paper would benefit greatly from a thorough description of “reference video” much earlier in the paper, as this is not a typical component of interpreting services, thus, was difficult to make sense of until arriving at the photo examples, and still not immediately.
Lastly, there are quite a few typographical errors and inconsistencies, particularly in numbering and spacing between numbering.
Author Response
In order to correct cultural misconceptions and inappropriate terminology, the authors reviewed related studies and modified them as follows.
(1) In other related studies, it was found that DHH was used more than PHL. So the authors changed all PHL to DHH.
(2) Also, about the use of sign language by country, the words ASL (American sign language) and BSL (British Sign Language) were used. So, in this paper, when the sign language is specifically used in the experiment, it is replaced with KSL (Korean Sign Language).
(3) “communication problem” (line 67…) : The paragraph has been rewritten.
(4) “hand gestures” (line 479….) : hands or hand shapes
(5) The authors added more references and modified this part to make it more understandable. Please check line 68~82.
(6) As the reviewer mentioned, the authors specifically indicated the sign language for each country.
(7) I am not sure that “easily” is the best term to describe accessibility of TV broadcasting to “everyone.” I would recommend replacement with “typically.” (Line 42.) : The part has been modified.
(8) Some reference-heavy claims are missing citations, such as lines 47-49; 52-65. : References are added.
(9) To quickly explain 'Reference videos', the following sentence is added in the abstract, 'which are mainly placed on the screen background,' after the Reference videos.ex) 'Reference videos which are mainly placed on the screen background, the second information delivery element, were considered less important to DHH participants compared to sign language interpreters and subtitles,~'
Reviewer 3 Report
The author has presented a design proposal for sign language services in TV broadcasting from the perspective of people with hearing loss. The current version of the paper is well organized and technically sound. The author should address the following comments for more improvement.
- Remove full stop(.) from the end of the title.
- It would be better to use 'sign people' and 'non-sign people, instead of using Deaf or Hard-of-Hearing (DHH) people and non-disabled people respectively.
- The author can include more diversity in the training and test cases including cluttered background with illumination variations, and Variations of the signers (male, female, different ages, different skin-colored, etc) following the paper which is presented below.
Rahaman, M.A., Jasim, M., Ali, M.H. et al. Bangla language modeling algorithm for automatic recognition of hand-sign-spelled Bangla sign language. Front. Comput. Sci. 14, 143302 (2020). https://doi.org/10.1007/s11704-018-7253-3
The conclusion should be more concise and informative highlighting contributions with a summary of the major analysis, limitations and future scopes, and real-life applications.
Author Response
1. Remove full stop(.) from the end of the title.
--> As the reviewer recommended, we have removed it.
2. It would be better to use 'sign people' and 'non-sign people, instead of using Deaf or Hard-of-Hearing (DHH) people and non-disabled people respectively.
-->We first used ‘People with hearing loss (PHL)’. When revising the manuscript, but in the first revision we have changed with ‘People who are Deaf or Hard-of-Hearing (DDH)’ as in many other related papers. The term 'sign people' that the reviewer suggested have been widely used in much broader meaning, even including Baby Sign Language and baseball sign. Since ‘sign people’ is a too broad term, we like to adhere to the term ‘People who are Deaf or Hard-of-Hearing (DDH)’, in this paper. Therefore, we hope the reviewer understand the use of this term.
3. The author can include more diversity in the training and test cases including cluttered background with illumination variations, and Variations of the signers (male, female, different ages, different skin-colored, etc) following the paper which is presented below.
--> As the reviewer suggested, it would be interesting to investigate how the factors may affect the experience of DHH people watching TV. In this manuscript, we have focused on the very fundamental elements such as layout and background of a sign language interpreter, and subtitle. We have mentioned the other factors for the future work.
4. The conclusion should be more concise and informative highlighting contributions with a summary of the major analysis, limitations and future scopes, and real-life applications.
--> As the reviewer recommended, we have restructured the conclusion to be more concise as in Section 5 and 6.
Round 2
Reviewer 1 Report
Presentation of the analysis in statistical description has not been improved that may have provided scientific validity of the results.
Author Response
Presentation of the analysis in statistical description has not been improved that may have provided scientific validity of the results.
-->As the reviewer mentioned, we have measured the Cronbach's alpha value for inter-scorer reliability to the questions to provide the scientific validity of the results (pp. 11, line 317-320).